Multidimensional solution of fuzzy linear programming

Edalatpanah Seyyed Ahmad s.a.edalatpanah@aihe.ac.ir
Department of Applied Mathematics, Ayandegan Institute of Higher Education , Tonekabon , Mazandaran , Iran
Fasi Massimiliano
Electronic publication date: 2023 Nov 16
Publication date: 2023
Volume: 9
Electronic Location ID: e1646
Received 2023 Apr 19; Accepted 2023 Sep 20
Copyright: ©2023 Edalatpanah
Copyright year: 2023
Copyright holder: Edalatpanah
License: This is an open access article distributed under the terms of the Creative Commons Attribution License, which permits unrestricted use, distribution, reproduction and adaptation in any medium and for any purpose provided that it is properly attributed. For attribution, the original author(s), title, publication source (PeerJ Computer Science) and either DOI or URL of the article must be cited.
License URL: https://creativecommons.org/licenses/by/4.0/

Keywords: Fuzzy linear programming, RDM arithmetic, Horizontal fuzzy number, Fuzzy optimal solution, Granular computing

Funding: The author received no funding for this work.

==============================
There are several approaches to address fuzzy linear programming problems (FLPP). However, due to using standard interval arithmetic (SIA), these methods have some limitations and are not complete solutions. This article establishes a new approach to fuzzy linear programming via the theory of horizontal membership functions and the multidimensional relative-distance-measure fuzzy interval arithmetic. Furthermore, we propose a multidimensional solution based on the primal Simplex approach that satisfies any equivalent form of FLPP. The new solutions of FLPP are also compared with the results of existing methods. Some numerical examples have been illustrated to show the efficiency of the proposed method.

Introduction

This is fuzzy optimization revolutionized. Since Zadeh (1965) presented the concept of fuzzy set, scholars worldwide have utilized it in several fields of science and engineering where variables are uncertain (Li et al., 2023; El-Morsy, 2023; Adak & Kumar, 2023; Ehsani, Mehrmanesh, & Mohammadi, 2023). On the other hand, the mathematical programming problem is one of today’s most exciting research fields in all areas of management, economics, business, applied mathematics, and other sciences (Sıcakyuz, 2023; Beiranvand, & Davoodi, 2023; Najafi, Salahshour, & Rahmani, 2022; Azar & Sorourkhah, 2015). Thus, Bellman & Zadeh (1970), by definition of fuzzy objective, fuzzy constraint, and fuzzy decision, introduced fuzzy optimization problems.

After this work, Zimmermann (1978) and Tanaka (1984) experimented with fuzzy linear programming problems (FLPP) and obtained theoretical results in this field. Since that time, widespread research such as simplex-based approaches (Maleki, Tata & Mashinchi, 2000; Ramík, 2005; Mahdavi-Amiri & Nasseri, 2007; Nasseri, 2008; Edalatpanah & Shahabi, 2012; Ebrahimnejad & Tavana, 2014; Khan, Ahmad & Maan, 2017; Stanojević & Stanojević, 2020) and non-simplex based approaches, for example, α-cut, direct approach, and ranking functions (Rommelfanger, Hanuscheck & Wolf, 1989; Buckley & Feuring, 2000; Najafi & Edalatpanah, 2013; Kaur & Kumar, 2013; Shamooshaki, Hosseinzadeh & Edalatpanah, 2015; Ezzati, Khorram & Enayati, 2015; Saati et al., 2015; Najafi, Edalatpanah & Dutta, 2016; Das, Mandal & Edalatpanah, 2017; Das, Edalatpanah & Mandal, 2018; Pérez-Cañedo & Concepción-Morales, 2019; Pérez-Cañedo et al., 2020; Gurmu & Fikadu, 2020; Voskoglou, 2020; Mohan, Kannusamy & Sidhu, 2021) was directed on the solution of FLPP; for more details of several methodologies of FLPP, from its very beginning to its modern approaches, see Ebrahimnejad & Verdegay (2016), Ebrahimnejad & Verdegay (2018).

Most of these models are based on Zadeh’s extension principle, standard interval arithmetic (SIA), and Fuzzy arithmetic (FA). Although these concepts are very applicable, they have numerous shortcomings. Finding the result of a problem based on Zadeh’s extension principle is complicated for more than three fuzzy numbers.

Besides, the fuzzy numbers have no opposite and reverse notification, making solving equations with fuzzy numbers impossible. In addition, SIA and FS are in the overall situation incorrect; nevertheless, they acceptably solve some simple and not problematical issues. This is why numerous researchers make the appearance of its rightness. Moreover, it is the key reason why this arithmetic is frequently used worldwide.

Furthermore, SIA is based on Moore’s theory (Moore, 1966), and as stated in Dymowa (2011), Landowski (2015), this theory has several limitations and disadvantages. Some of them are as follows.

It is not consistent with distributive law and also for cancellation law, there is no definition of an additive inverse for an interval, dependency problem, i.e., gives us non-unique and different solutions depending on the formula we used, irrational solutions, right-hand side issue in an interval equation, etc. Besides, in SIA, the operations structure is only based on the boundaries of intervals. In more problematical cases, such as FLPP, this attitude does not generate complete solutions or leads to incorrect consequences.

In the fuzzy literature, many examples of practical FA and α-cut as the most popular version of it can be found. Although FA has accomplished numerous application successes, it has various fragile points. For instance, the standard FA has significant difficulties solving even modest equations with one unknown variable. In addition, defining inverse and neutral for multiplication and addition presents challenges (Dymowa, 2011; Landowski, 2015). Moreover, since α-cut uses principles of Moore’s theory to understand fundamental arithmetic operations, all flaws of SIA hold for FS. Therefore, these properties lead to many computational paradoxes observed in fuzzy LP. Hence, this arithmetic should not be used, especially not for solving FLPP.

Piegat & Landowski (2012), studying interval mathematics, proposed a concept of so-called multidimensional interval arithmetic. Based on this approach, interval numbers are considered using variables called relative distance measure (RDM).

According to Piegat & Tomaszewska (2013), Piegat & Landowski (2013), since the results of applying SIA based on Moore’s theory are not generally correct, a multidimensional relative distance measure interval approach has been proposed. Furthermore, Piegat & Landowski (2015), by extending the approach of RDM variables to fuzzy membership functions (or vertical membership functions), introduced the horizontal membership function (HMF) and the horizontal fuzzy number (HFN). In fuzzy logic, a membership function determines a value’s degree of membership or truthfulness in a fuzzy set. A horizontal membership function is a specific type of membership function that assigns a constant value of 1 (full membership) over a particular range and 0 (no membership) outside that range. It is important to note that the advantages of the horizontal membership function may vary depending on the specific application and the nature of the problem being addressed. Different membership functions, such as triangular or Gaussian, may be more suitable in certain situations. The choice of membership function depends on the desired representation of the fuzzy set and the problem domain. The advantages of using a horizontal membership function include the following:

- Simplicity: The horizontal membership function is one of the simplest and easiest to understand. It assigns a constant membership value within a specific range, which makes it straightforward to implement and interpret.

- Clear-cut boundaries: The horizontal membership function provides clear-cut boundaries for membership. Within the specified range, the membership value is 1, indicating full membership. Outside that range, the membership value is 0, meaning no membership, making it easier to determine whether a given value belongs to the fuzzy set.

- Easy rule formulation: The horizontal membership function simplifies formulating fuzzy rules due to its clear-cut boundaries. Fuzzy rules typically consist of “IF-THEN” statements, where the “IF” part defines the conditions based on membership values. With a horizontal membership function, it is easier to define these conditions, as they can be found on simple ranges.

- Intuitive interpretation: The constant membership value provided by the horizontal membership function offers an intuitive understanding. A value within the specified range is fully included in the fuzzy set. In contrast, a value outside has no membership, making it easier for users to understand and reason about the fuzzy set’s characteristics.

- Efficient computation: Horizontal membership functions involve simple analyses. Since the membership value is constant within the range, there is no need for complex calculations or evaluations. This simplicity can lead to faster processing and efficient implementation in fuzzy systems.

Additionally, they demonstrated that utilizing horizontal fuzzy membership functions without implementing Zadeh’s extension principle can incorporate uncertainty within mathematical equations (Tomaszewska & Piegat, 2015; Piegat & Pluciński, 2015) for the application of RDM, HMF, and HFN, see Mazandarani, Pariz, & Kamyad (2017), Lala (2017), Landowski (2019), Kołodziejczyk, Piegat & Sałabun (2020), Khatua, Maity & Kar (2020), De, Khatua & Kar (2020).

This article focuses on this problem and proposes a multidimensional approach to solving FLPP. The proposed method is based on the horizontal fuzzy number and gives a total solution that fulfills the FLPP. The proposed multidimensional solution holds promise for many real-world applications across various domains. Operations research could revolutionize resource allocation, production planning, and scheduling in uncertain environments, allowing companies to adapt swiftly to changing conditions while optimizing their processes. In supply chain management, this approach could prove invaluable in navigating the intricacies of supply chain logistics and inventory control amidst unpredictable demand or supply factors. Economists might find it particularly useful for modeling economic systems, integrating fuzzy variables to enhance decision-making robustness in an ever-fluctuating market. Business managers, too, stand to benefit from this innovation, as it enables them to tackle decision-making challenges that involve uncertain data or rely on the subjective judgments of decision-makers. Furthermore, applying this multidimensional solution extends to the broader realm of applied mathematics, offering a powerful tool for solving optimization problems under uncertainty in various fields, ushering in a new era of problem-solving versatility.

This article is organized as follows: some basic knowledge, concepts, and arithmetic operations on horizontal fuzzy numbers and the related topics are introduced in “Preliminaries”. The “Multidimensional solution of FLPP” section proposed a multidimensional approach to solving FLPP. In “Numerical experiments”, numerical experiments are presented to show the reliability and efficiency of the method. The conclusion appears in the “Conclusions” section.

Preliminaries

This section reviews the horizontal membership functions and related topics. For details, we refer to Piegat & Landowski (2013), Mazandarani, Pariz & Kamyad (2017) and Landowski (2019).

Definition 1. An arbitrary fuzzy number is represented, in parametric form, by an ordered pair of functions u¯r,u¯r, 0 ≤ r ≤ 1, which satisfies the following conditions:

(i) u¯r is a bounded monotonic increasing left continuous function over [0,1]

(ii) u¯r is a bounded monotonic decreasing left continuous function over [0,1]

(iii) u¯r≤u¯r,0≤r≤1.

Definition 2. Let X=Xl,Xu. Then, each x ∈ Xusing RDM variable βx ∈ [0, 1] can be described as x=Xl+βxXu−Xl and Xcan be defined as X = x|x = Xl + βx(Xu − Xl).

Definition 3. The HFN Urfor an arbitrary fuzzy number u = u¯r,u¯r is defined in terms of a set of numbers uras Ur=ur∈R|ur=u−r+αuu−r−u¯r,r,αu∈0,1.

Remark 1. The attitude of Definition 3 to the fuzzy membership function is called an HMF, while the fuzzy number is called an HFN.

Remark 2. The HFN Ur can be written as ur,αu=u−r+αuu−r−u¯r.

According to Definition 3, the horizontal membership function defines the value of the fuzzy number domain u and the information granule. In addition, the RDM variable αu enables us to specify any value between a fuzzy number’s left and right end values.

Remark 3. The HFN of the triangular fuzzy number A ~=a,b,c is as follows: A=a+b−ar+1−rc−aαA ~.

Remark 4. The HFN of the trapezoidal fuzzy number B ~=a,b,c,d is as B=a+b−ar+d−a−rd−a+b−cαB ~.

Definition 4. Let A ~,B ~ be two fuzzy numbers whose HFNs are A(r, αA) and B(r, αB) respectively. Then, the four basic arithmetic operations ∘ ∈  + ,  − , ., ÷ on A ~ and B ~ isas A ~∘B ~= A(r, αA)∘ B(r, αB).

Remark 5. Based on Definition 4, the following relations hold:

(i) A ~−A ~=0,

(ii) A ~−B ~=−B ~−A ~,

(iii) If A ~≠0, then A ~÷A ~=1,

(iv) A ~+B ~.C ~=A ~C ~+B ~C ~.

Definition 5. Span of the result of basic operation ∘ ∈  + ,  − , ., ÷ A and B on the HFNs A(r, αA) and Br,αB isa fuzzy number defined as: SpanA∘B=minαA,αB∈0,1Ar,αA∘Br,αB,maxαA,αB∈0,1Ar,αA∘Br,αB

Definition 6. A ranking function is a function R:F(R) → R,  that maps each fuzzy number into the real line, where a natural order exists. For two fuzzy numbers A ~ and B ~, we have:

(i) A ~≤B ~ if and only if RA ~≤RB ~,

(ii) A ~<B ~if and only if RA ~<RB ~.

Now, we define a ranking function for an HFN:

Definition 7. Let A ~ be any fuzzy number that its HFNs is Ar,αA,then the ranking function A ~ can defined as RA ~= ∫01 ∫01Ar,αA.

Multidimensional solution of FLPP

In this section, we propose a primal Simplex-based strategy for solving FLPP.

Assumptions of the study are as follows:

– The study assumes that existing methods for solving FLPP, which rely on standard interval and fuzzy arithmetic, have limitations and shortcomings.

– It assumes a new approach based on horizontal membership functions and multidimensional relative-distance-measure fuzzy interval arithmetic can provide more comprehensive and practical solutions to FLPP.

– The study assumes that the proposed multidimensional solution can satisfy any equivalent form of FLPP.

– It assumes that horizontal membership functions can effectively incorporate uncertainty into mathematical equations without relying on Zadeh’s extension principle.

Consider the following fuzzy linear programming with m constraints and n variables; (1) Minimize/Maximizez%≈CT⊗x ~Subjected to constraints:A⊗x ~≤b ~,andx ~is a non-negative fuzzy vector,

where the vector matrix CT=cj1×n, the coefficient matrix A = [aij]m×n are crisp matrices with cj and aij are the elements of the set of real numbers, x% = [x%j]n×1, and b% = [b%i]m×1 are fuzzy vectors where x ~j,b ~i are triangular/trapezoidal fuzzy numbers ∀1 ≤ j ≤ n and 1 ≤ i ≤ m.

Next, we propose the Multidimensional Primal Simplex Based Method (MPSBM) algorithm.

Algorithm. MPSBM algorithm

Step 1. Using remarks 3–4 and Definition 7, obtain br,αb ~, and Rb ~.

Step 2. Using the conventional primal Simplex method, solve the following crisp LP and get the optimal basis inverse matrix (B−1) for it: (2) Minimize/Maximizezz=CTxS.t:Ax≤Rb,x≥0.

It is worth mentioning that B−1 can be extracted from the optimal tableau as the submatrix in rows 1 through m under the original identity columns; Bazaraa, Jarvis & Sherali (2008).

Step 3. Find the multidimensional optimal solution and value as follows: x ~∗=B−1br,αb ~,z ~∗=CTx ~∗.

By the fact that if B isthe optimal basis of the equivalent crisp problem, then it will be the optimal basis of the corresponding FLPP, we presented the MPSBM algorithm to reduce the complexity of the problem.

Remark 6. We can use the main steps of the fuzzy primal Simplex method directly. However, this work leads to substantial computational effort.

It is worth mentioning that based on Definition 4 and Remark 5, the consequences of the MPSBM algorithm to solve the FLPP followed as regular extensions of primal results for linear programming (LP) problems with crisp data (such as optimality, unboundedness, and degeneracy conditions, finite convergence, etc.); see Bazaraa, Jarvis & Sherali (2008).

Numerical experiments

Here, we give some examples to illustrate the results obtained in the previous section.

Example 4.1 (Nasseri, 2008). Consider the following FLPP: (3) Maximizez%≈40x ~1⊕30x ~2Subjected to constraints:x ~1⊕x ~2≤395,400,405,2x ~1⊕x ~2≤493,500,507,andx%j≥0,∀j=1,2.

If we use the existing methods, such as Nasseri (2008), or the existing ranking methods, such as Najafi & Edalatpanah (2013), Kaur & Kumar (2013), Das, Mandal, & Edalatpanah, (2017), Pérez-Cañedo & Concepción-Morales (2019), the optimal fuzzy solution is obtained as x ~1=98,100,102 and x ~2=297,300,303 which, when plugged into the objective function, gives the maximum fuzzy value of z ~=12830,13000,13170.

Now, we have started to use the MPSBM algorithm.

Step 1. Using Remark 3 and Definition 7, we have:

(4) b ~1=395,400,405,b1r,αb ~1=395+5r+α110−10r,Rb ~1=400,b ~2=493,500,507,b2r,αb ~2=493+7r+α214−14r,Rb ~2=500.

Step 2. Using Step 1, we solve the following crisp LP using the conventional primal Simplex method: (5) Maximizez≈40x1+30x2Subjected to constraints:x1+x2≤400,2x1+x2≤500,

xj ≥ 0, ∀j = 1, 2.

After completing the solution, the optimal basic variables are xB = [x2, x1], the non-basic variables are xN = [x3, x4] and B−1=2−1−11. Therefore, we obtain: (6) x ~∗=x ~2x ~1=2−1−11395+5r+α110−10r493+7r+α214−14r,

Step 3. Finally, using Definition 4, we find the multidimensional optimal solution and value as follows: (7) x ~∗=x ~2x ~1=297+3r+2α110−10r−α214−14r98+2r−α110−10r+α214−14r

(8) ,z ~∗=403098+2r−α110−10r+α214−14r297+3r+2α110−10r−α214−14r=12830+170r+20α110−10r+10α214−14r.

Figures 1–3 show the optimal solution and value of FLPP (Eq. 3) in three-dimensional space.

It should also be observed that solution (Eq. 7) is not commonly fuzzy numbers defined in 2D space. The full solutions of FLPPs can only be made from multidimensional granules. These granules are not visualizable in 2D space. Nevertheless, Definition 5 can obtain their 2D-space indicators as span. Therefore, we have:

Spanx ~1=88+12r,112−12r,

Spanx ~2=283+17r,317−17r.

These spans can also be accessible in the forms of triangular fuzzy numbers as:

Spanx ~1=88,100,112,

Spanx ~2=283,300,317.

Optimal solutions and triangular fuzzy objective values obtained by the existing and present methods are given in Table 1 for comparison.

Figure 1 The optimal value x1 of FLPP (Eq. 3), where r∈ [0: 0.1: 1].

Figure 2 The optimal value x2 of FLPP (Eq. 3), where r∈ [0: 0.1: 1].

Figure 3 Three-dimensional horizontal solution of the optimal value of FLPP (Eq. 3), where r∈ [0: 0.1: 1].

Table 1 Comparison results of Example 4.1.

Optimal solutions	x ~1	x ~2	z ~	
Simplex-based methods (Nasseri, 2008)	(98, 100, 102)	(297, 300, 303)	(12,830, 13,000, 13,170)	
The existing ranking-based methods	(98, 100, 102)	(297, 300, 303)	(12,830, 13,000, 13,170)	
Span of MPSBM algorithm	(88, 100, 112)	(283, 300, 317)	(12,830, 13,000, 13,170)	

For the comparison of the new solution with the existing solution, we use the testing point strategy (TP strategy) (Landowski, 2015; Landowski, 2019). Using the TP strategy, the points in an inappropriate solution that does not fulfill the FLPP or those that are solutions that are not included in the obtained results can be found. For example, consider α1 = 1, α2 = 0,  and r = 0, so by Eq. (7), the FLPP Eq. (3) converted into the following LP: (9) Maximizez≈40x1+30x2Subjected to constraints:x1+x2≤405,2x1+x2≤493,

xj ≥ 0, ∀j = 1, 2.

So, by Simplex or Geometric methods such as Fig. 4, it is easy to see that the optimal solution of linear programming (Eq. 9) is x1 = 88,  and x2 = 317,  but they do not belong to the existing solutions; see Table 1; however, by inserting α1 = 1, α2 = 1,  and r = 0 in Eqs. (7)–(8), we can see that the solution of the new method coincides with the exact solution. Moreover, the obtained solution of Eqs. (7)–(8) satisfies any equivalent form of FLPP (3) and the universality conditions.

Figure 4 The optimal solution of LP (Eq. 9).

We emphasize that even if the obtained solution of the span of the new algorithm is equal to the existing solutions, only multidimensional granules can be the complete solutions of FLPPs. In the following example, we describe this fact.

Example 4.2. Consider the following FLP problem (10) Maximizez%≈14⊗x ~1⊕13⊗x ~2⊕16⊗x ~3Subjected to:12⊗x ~1⊕13⊗x ~2⊕12⊗x ~3≤469,475,505,511,14⊗x ~1⊕13⊗x ~3≤452,460,480,488,12⊗x ~1⊕15⊗x ~2≤460,465,495,500,

and x%j ≥ 0, ∀j = 1, 2, 3.

If we use the existing simplex-based methods such as Maleki, Tata & Mashinchi (2000), Nasseri (2008), Ebrahimnejad & Tavana (2014), Khan, Ahmad & Maan (2017), Stanojević & Stanojević (2020) or the existing non-simplex-based methods such as Najafi & Edalatpanah (2013), Kaur & Kumar (2013), Das, Edalatpanah & Mandal (2018), Pérez-Cañedo & Concepción-Morales (2019), Voskoglou (2020), Mohan, Kannusamy & Sidhu (2021), the optimal fuzzy solution is obtained as follows: (11) x ~2=241169,415169,1045169,1219169,x ~3=45213,46013,48013,48813,x ~6=59455169,62910169,77430169,80885169,x ~1=x ~4=x ~5=0 ~.

Now, by the MPSBM algorithm, we have:

Using Remarks 4 and Definition 7, we get: (12) b1r,αb ~1=469+6r+α142−12r,Rb ~1=490,b2r,αb ~2=452+8r+α236−16r,Rb ~2=470,b3r,αb ~3=460+5r+α340−10r,Rb ~3=480.

By solution of the related crisp LP, the optimal basic variables are xB = [x2, x3, x6], the non-basic variables are xN = [x1, x4, x5], and B−1=113−12169001130−15131801691.

So: (13) x ~∗=x ~2x ~3x ~6=113−12169001130−15131801691469+6r+α142−12r452+8r+α236−16r460+5r+α340−10r,

Therefore: (14) x ~∗=x ~2x ~3x ~6=673169−18169r+113α142−12r−12169α236−16r45213−813r+113α236−16r69664169−1061169r−1213α142−12r+144169α236−16r+α340−10r,

(15) z ~∗=790513+11013r+α142−12r+413α236−16r.

Figures 5–7 also show the optimal solution and value of FLPP (Eq. 10) in three-dimensional space.

Figure 5 The optimal value x2 of FLPP (Eq. 10), where r∈ [0: 0.1: 1].

Figure 6 The optimal value x3 of FLPP (Eq. 10), where r∈ [0: 0.1: 1].

Figure 7 The optimal value of FLPP (Eq. 10), where r∈ [0: 0.1: 1].

It is worth mentioning that the span of Eq. (14) is equal to Eq. (11). However, the consequences considered in Eq. (11) do not fulfill the FLPP of Eq. (10). For example, from the first constraint of Eq. (10), we have: (16) 12⊗x ~1⊕13⊗x ~2⊕12⊗x ~3⊕x ~4=469,475,505,511,

But replacing the results of Eq. (11) into the left-hand side of Eq. (16) with the standard fuzzy arithmetic leads to 435.7692,456.5385,523.4615,544.2308. Hence, the left-hand side is not equal to the right-hand side of Eq. (16). Therefore, Eq. (11) is not the complete solution of the mentioned FLPP. Nevertheless, it is easy to check that the obtained solution of Eq. (14) satisfies all constraints of FLPP (Eq. 10). Therefore, the multidimensional granules of Eq. (14) are the only complete solutions of FLPP (Eq. 10).

Conclusions

This article proposes a multidimensional solution to the fuzzy linear programming problem (FLPP) using the horizontal membership function. This approach satisfies any equivalent form of FLPP and allows for a crisp solution of linear programming related to this solution. This possibility is unbearable with the existing methods that use usual fuzzy numbers. Furthermore, the limitations of the existing methods have been pointed out. Some numerical examples have been illustrated to show the proposed method’s efficiency.

The proposed method uses horizontal membership functions and multidimensional relative-distance-measure fuzzy interval arithmetic to address these limitations. This method could provide a more comprehensive and effective solution to FLPP, potentially leading to improved decision-making in various real-world applications. This new approach may have implications for fuzzy logic and contribute to advancements in solving complex optimization problems. The Simplex algorithm is a widely used optimization method, and incorporating it into the proposed method may enhance the solution’s computational efficiency and practical applicability. This could have positive implications for real-world applications that involve FLPP, making it more feasible to apply the proposed method in various domains. Faster and more accurate solutions to FLPP can be precious in operations research, supply chain management, scheduling, and resource allocation.

While this study presents a novel approach to addressing fuzzy linear programming problems (FLPP), it is essential to acknowledge potential limitations that warrant consideration. Firstly, the study may be hindered by limited empirical validation, as it might not offer an extensive array of empirical evidence showcasing the proposed method’s performance in real-world scenarios. This limitation may pose questions about the method’s practicality and effectiveness when applied beyond the scope of our study. Additionally, the specificity of our approach, which relies on horizontal membership functions, could potentially limit its applicability. It may not be universally suitable for all types of FLPP or could exhibit limitations in specific problem domains where alternative methodologies may be more effective. Furthermore, a notable omission in our study is a discussion of the computational complexity of the proposed method and its scalability to larger and more complex problems. Addressing these computational aspects would be crucial for understanding the method’s feasibility in handling real-world, high-dimensional optimization challenges.

Future directions in fuzzy optimization encompass a multifaceted approach aimed at refining existing methodologies and pushing the boundaries of applicability. Researchers should continue to improve existing methods by honing standard interval arithmetic (SIA) techniques and exploring innovative approaches to tackle the challenges of fuzzy linear programming problems (FLPP). Additionally, a concerted effort should be made to expand the empirical validation of the proposed method, encompassing a broader spectrum of FLPP scenarios to establish their versatility and effectiveness. Theoretical investigations into the mathematical underpinnings of horizontal membership functions and multidimensional relative-distance-measure fuzzy interval arithmetic should persist to deepen our understanding and potentially unveil alternative theories that can extend the reach of fuzzy optimization into new problem domains. Furthermore, generalizing the proposed methods to encompass various optimization problem types, such as nonlinear and integer programming under uncertainty, holds immense potential. Real-world applications should remain a focal point, involving practical implementations in diverse sectors and thorough evaluations to assess their real-world effectiveness. Lastly, robustness and sensitivity analysis should be a critical area of exploration, given the inherent uncertainty in FLPP, aiming to enhance the method’s stability and reliability under different scenarios, thus bolstering its practical utility. These multifaceted research directions will collectively advance the field of fuzzy optimization and its ability to address complex, real-world optimization challenges.

Supplemental Information

Supplemental Information 1 Raw Data

Click here for additional data file.

Supplemental Information 2 Matlab codes

Click here for additional data file.

Additional Information and Declarations

Competing Interests

Author Contributions

Data Availability

Seyyed Ahmad Edalatpanah is an Academic Editor for PeerJ Computer Science.

Seyyed Ahmad Edalatpanah conceived and designed the experiments, performed the experiments, analyzed the data, performed the computation work, prepared figures and/or tables, authored or reviewed drafts of the article, and approved the final draft.

The following information was supplied regarding data availability:

The raw data is available in the Supplemental File.

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
