# Peer review of "Multidimensional solution of fuzzy linear programming"

_PeerJ Computer Science, doi:10.7717/peerj-cs.1646_

## Round 0.1 · original submission · Major Revisions

Please revise your paper according to the reviewer's comments.
Thank you

Reviewer 1 ·

Basic reporting

This paper proposes a new approach to solving fuzzy linear programming problems (FLPP) based on the theory of horizontal membership functions and multidimensional relative-distance-measure fuzzy interval arithmetic. The proposed method is a multidimensional solution based on the primal Simplex approach and is compared with existing methods. The paper is well-organized and easy to follow. The following are some comments with minor revision:

Strengths:

 The paper provides a comprehensive review of previous research on FLPP, which makes it easier for readers to understand the context and significance of the proposed method.
 The paper proposes a new approach that overcomes the limitations of existing methods and provides a complete solution to FLPP.
 The numerical examples presented in the paper illustrate the efficiency of the proposed method.
 The paper is well-written and well-organized.
Minor Revision:
• It would be helpful to provide a clearer explanation of the horizontal membership function and its advantages over other approaches.
• The paper could benefit from providing more detailed information on the implementation of the proposed method, including any software or tools used.
• Lines 26 and 27 are empty
• Add some recent references in literature review.
• Check Definition 3 once again.
• Check remarks 3 and 4 once again.
• Check some formulas, especially fuzzy symbols.
• The paper could be improved by providing more information on the significance and potential impact of the proposed method on real-world applications.
• explain future suggestions in the conclusion section.
• Update the references

Experimental design

.

Validity of the findings

.

Additional comments

.

Cite this review as

Reviewer 2 ·

Basic reporting

The author has stated the research problem well, reviewed the background, proposed an effective approach, and has shown the effectiveness of the proposed approach compared to previous studies by presenting numerical examples. The manuscript has a high scientific standard, and I believe it can attract the attention of the journal's audience. Therefore, I have no practical comment.

Experimental design

The author has presented a novel approach to fuzzy LP via horizontal membership functions theory and the multidimensional relative-distance-measure fuzzy interval arithmetic, as well as a multidimensional solution based on the primal Simplex approach satisfying every equivalent FLPP form. The author has demonstrated the proposed method's efficiency by comparing the existing methods' results through some numerical examples.

Validity of the findings

no comment

Additional comments

I suggest a few considerations to improve the quality of work:
1- Some minor grammatical errors (lines 11, 14, etc.)
2- It is better to show that the literature supports the second and third sentences in the introduction section (lines 20 to 24).
3- Determine if there is any content in lines 26 and 27.
4- In lines 47 to 49, a sentence is repeated twice.
5- Please check the symbols and indices carefully once again (for example, lines 103, 109, 117, etc.)
6- From line 114 onwards, many signs have changed (of course, it may be due to the conversion of Word to PDF). Please review these items and correct them if necessary.
7- Correct the referencing style in lines 181 and 182.
8- What do the numbers in brackets at the beginning of line 232 mean? Is it a reference to previous works?
9- Can you provide some directions for future research in the conclusion section?

Cite this review as

---

## Round 0.2 · Major Revisions

Thank you for submitting the revised manuscript. The previous Academic Editor is not available and so I have taken over handling it.

As you will see, a number of issues with the current version have been identified.

Please revise your paper according to the comments of referees 3 and 4, with particular attention to the wrong mathematical symbols and the grammatical issues.

**Language Note:** The Academic Editor has identified that the English language must be improved. PeerJ can provide language editing services - please contact us at copyediting@peerj.com for pricing (be sure to provide your manuscript number and title). Alternatively, you should make your own arrangements to improve the language quality and provide details in your response letter. – PeerJ Staff

Reviewer 1 ·

Basic reporting

The paper has ben improved taking in consider all suggestions.

Experimental design

no comment

Validity of the findings

no comment

Additional comments

no comment

Cite this review as

Reviewer 2 ·

Basic reporting

The paper can be accepted in the present form.

Experimental design

Appropriate.

Validity of the findings

Appropriate.

Additional comments

None.

Cite this review as

Reviewer 3 ·

Basic reporting

1.. It is not possible that the research work to be done with 'no assumptions' - this has to be addressed
broadly check spelling and grammar
2.More discussion on real-world application problems in the literature has not been addressed

Experimental design

1.Although future works are proposed, the limitations of the present work remains unclear (the research assumptions will help provide clarity to the limitations of the work). The content of this paper need to be significantly improved before publication

Validity of the findings

1.Error analysis on the results/findings to be done.

Additional comments

no comments

Cite this review as

Reviewer 4 ·

Basic reporting

The novelty of paper is good. It matches with aims and scope of the journal. However, some grammatical mistakes can be improved.

Experimental design

The experimental design presented in manuscript is relevant and meaningful.

Validity of the findings

The comparison of proposed method is given the the existing ones. But it should be explained that in comparison table the optimal value of proposed method and the existing methods, is same. Then how it will be better?

Additional comments

Comment 1: Mathematical symbols are not correct in PDF file (In definition 4, remark 5, definition 6, equation no 1 and at many other places). Kindly recheck the symbols at all places.
Comment 2: There are some grammatical mistakes in the manuscript. Kindly rectify these.
Comment 3: What is FFLP? Where it is defined?
Comment 4: What is HFM and HMF ? Write its full form where it is first time used.
Comment 5: Open bracket is missing in remark 2.
Comment 6: How fuzzy variable x is converted into crisp value in Step 2 of Algorithm?
Comment 7: Future scope should be given in a single paragraph, not in points. Kindly rewrite in precise form.
Comment 8: In references, some journal names are not in same font size, even some are in “Italic” others are not.
Comment 9: Some journal’s names are in short form. Kindly follow the same format for all references.

Cite this review as

---

## Round 0.3 · accepted · Accept

Both reviewers have confirmed that all their concerns have been addressed. Therefore, the manuscript can be accepted for publication.

Reviewer 3 ·

Basic reporting

All the points suggested have been addressed, In my opinion the article may be considered for publication

Experimental design

No comments

Validity of the findings

No comments

Additional comments

No comments

Cite this review as

Reviewer 4 ·

Basic reporting

As per the suggestions, the article is received in the revised form. All comments has been incorporated.
So, the Article may be accepted.

Experimental design

"No commnet"

Validity of the findings

"No comment"

Additional comments

"No comment"

Cite this review as